# Gradient-based Editing of Memory Examples for Online Task-free Continual Learning

**Xisen Jin    Arka Sadhu    Junyi Du    Xiang Ren**
University of Southern California
{xisenjin, asadhu, junyidu, xiangren@usc.edu}

## Abstract

We explore task-free continual learning (CL), in which a model is trained to avoid catastrophic forgetting in the absence of explicit task boundaries or identities. Among many efforts on task-free CL, a notable family of approaches are memory-based that store and replay a subset of training examples. However, the utility of stored seen examples may diminish over time since CL models are continually updated. Here, we propose Gradient based Memory EDiting (GMED), a framework for editing stored examples in continuous input space via gradient updates, in order to create more "challenging" examples for replay. GMED-edited examples remain similar to their unedited forms, but can yield increased loss in the upcoming model updates, thereby making the future replays more effective in overcoming catastrophic forgetting. By construction, GMED can be seamlessly applied in conjunction with other memory-based CL algorithms to bring further improvement. Experiments validate the effectiveness of GMED, and our best method significantly outperforms baselines and previous state-of-the-art on five out of six datasets[1].

## 1   Introduction

Learning from a continuous stream of data – referred to as *continual learning (CL)* or *lifelong learning* – has recently seen a surge in interest, and many works have proposed ways to mitigate CL models' catastrophic forgetting of previously learned knowledge [20, 32, 33]. Here, we study online *task-free* CL [3], where task identifiers and boundaries are absent from the data stream. This setting reflects many real-world data streams [6, 25] and offers a challenging testbed for online CL research.

Memory-based methods, a prominent class of approaches used for task-free continual learning, store a small number of training examples (from the data stream) in a memory and replay them at the later training iterations [33, 34]. Existing methods operate over the original examples in the data-stream and focus on identifying samples to populate the memory [4, 9] and finding samples in the memory to be replayed [2]. However, for continually updating models, using stored-seen examples in their original form, may lead to diminishing utility over time — i.e., model may gradually memorize the stored examples after runs of replay, as the memory refreshes slowly. An alternate approach is to use generative models to create samples that suffers more from forgetting such as in GEN-MIR in [2]. In practice, training the generator network with limited data is challenging and leads to low-quality generated examples. Further, in the online learning setup, the generative model itself suffers from forgetting. As such, generative models perform worse than their memory counter-parts.

In this paper, we present a novel memory-based CL framework, Gradient based Memory EDiting (**GMED**), which looks to directly "edit" (via a small gradient update) examples stored in the replay memory. These edited examples are stored (replacing their unedited counterparts), replayed, and further edited, thereby making the future replays more effective in overcoming catastrophic forgetting.

---

[1]Code can be found at `https://github.com/INK-USC/GMED`.

35th Conference on Neural Information Processing Systems (NeurIPS 2021).

Since no explicit generative model is involved, GMED approach retains the advantages of memory-based methods and is straightforward to train only inducing a small computation overhead.

The main consideration in allowing "editing" via a gradient update is the choice of the optimization objective. In light of recent work on designing alternative replay strategies [2, 7, 40], we hypothesize that "interfering" examples (*i.e.*, past examples that suffer from increased loss) should be prioritized for replay. For a particular stored example, GMED finds a small update over the example ("edit") such that the resulting edited example yields the most increase in loss when replayed. GMED additionally penalizes the loss increase in the edited example to enforce the proximity of the edited example to the original sample, so that the edited examples stay in-distribution. As a result, replaying these edited examples is more effective in overcoming catastrophic forgetting. Since GMED focuses only on editing the stored examples, by construction, GMED is modular, *i.e.*, it can be seamlessly integrated with other state-of-the-art memory-based replay methods [2, 5, 26].

We demonstrate the effectiveness of GMED with a comprehensive set of experiments over six benchmark datasets. In general, combining GMED with existing memory-based approaches results in consistent and statistically significant improvements with our single best method establishing a new state-of-art performance on five datasets. Our ablative investigations reveal that the gains realized by GMED are significantly larger than those obtained from regularization effects in random perturbation, and can be accumulated upon data augmentation to further improve performance.

To summarize, our contributions are two-fold: (i) we introduce GMED, a modular framework for task-free online continual learning, to edit stored examples and make them more effective in alleviating catastrophic forgetting (ii) we perform intensive set of experiments to test the performance of GMED under various datasets, parameter setups (*e.g.*, memory size) and competing baseline objectives.

## 2 Related Work

**Continual Learning** studies the problem of learning from a data stream with changing data distributions over time [20, 23]. A major bottleneck towards this goal is the phenomenon of catastrophic forgetting [33] where the model "forgets" knowledge learned from past examples when exposed to new ones. To mitigate this effect, a wide variety of approaches have been investigated such as adding regularization [1, 19, 30, 43], separating parameters for previous and new data [24, 36, 37], replaying examples from memory or a generative model [26, 33, 38], meta-learning [18]. In this work, we build on memory-based approaches which have been more successful in the online task-free continual learning setting that we study.

**Online Task-free Continual Learning** [3] is a specific formulation of the continual learning where the task boundaries and identities are not available to the model. Due to its broader applicability to real-world data-streams, a number of algorithms have been adapted to the task-free setup [2, 14, 15, 22, 44]. In particular, memory-based CL algorithms which store a subset of examples and later replay them during training, have seen elevated success in the task-free setting. Improvements in this space have focused on: *storing diverse examples* as in Gradient-based Sample Selection (GSS) [4], and *replaying examples with larger estimated "interference"* as in Maximally Interfered Retrieval (MIR) with experience replay [2]. In contrast , GMED is used in conjunction with memory-based approaches and explicitly searches for an edited example which is optimized to be more "effective" for replay.

**Replay Example Construction.** Moving away from replaying real examples, a line of works on *deep generative replay* [17, 35, 38] generates synthetic examples to replay with a generative model trained online. GEN-MIR [2] is further trained to generate examples that would suffer more from interference for replay. However, training a generative network is challenging, even more so in the online continual learning setup where the streaming examples are encountered only once leading to poorly generated examples. Moreover, the forgetting of generative networks themselves cannot be perfectly mitigated. As a result, these methods generally perform worse than their memory-based counter-parts. Instead of generating a new example via a generator network, GMED uses gradient updates to directly edit the stored example thereby retaining advantages of memory-based techniques while creating new samples.

Novel examples can also be constructed via *data augmentation* (*e.g.* random crop, horizontal flip) to help reduce over-fitting over the small replay memory [5]. Unlike GMED, these data-augmentations are usually pre-defined and cannot adapt to the learning pattern of the model. Constructing edited examples has also been used for adversarial robustness [31, 39]. The key difference lies in the

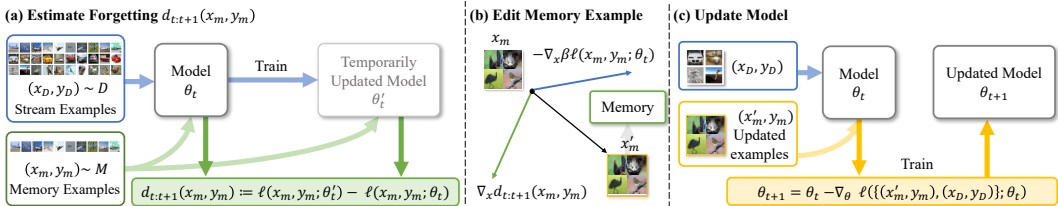

*Figure 1:* **A schematic of GMED framework**. (a) Given an example from the data-stream $(x_D, y_D)$ at time $t$, the model randomly draws an example from the memory $(x_m, y_m)$ and estimates "interference" after one-step roll-out (Eq. 2). (b) The example drawn from the memory is then updated using our proposed editing objective (Eq. 3) via gradient ascent, resulting in $(\hat{x}_m, y_m)$, and written back into the memory. (c) Finally, the model is updated using the edited example $(\hat{x}_m, y_m)$ and the example from the data-stream $(x_D, y_D)$ (Eq. 4).

optimization objective: while adversarial construction focuses on finding mis-classified examples [13, 28], GMED aims at constructing "interfered" examples that would suffer from loss increase in future training steps.

## 3 Method

We introduce the formulation of online task-free continual learning (Sec. 3.1) and then present our gradient-based memory editing method GMED (Sec. 3.2) and detail how GMED can be integrated with experience replay (Sec 3.3) and other memory-based CL algorithms (Sec. 3.4).

### 3.1 Problem Formulation

In continual learning (CL), we consider a (potentially infinite) stream $D$ of labeled examples $(x, y)$, having a non-stationary data distribution – *i.e.*, the data distribution $p(x, y)$ evolves over time. Let $(x_t, y_t)$ denote the labeled example (or a mini-batch of examples) received by the model at time-step $t$ in the data-stream $D$. We assume, for simplicity, that $(x_t, y_t)$ is generated by first sampling a *latent* "task" $z \sim p(z; t)$, followed by sampling a data example from a conditional data distribution $p(x, y|z)$, – *i.e.*, $(x_t, y_t) \sim p(x, y|z)$. Here $p(z; t)$ is non-i.i.d and time-dependent. In *task-free* CL setting, the latent task $z$ is *not* revealed to the model. Finally, we emphasize that while models in task-aware setup proceed to the next task only after convergence on the current task [19, 43], this work focuses on ***online** task-free CL* setup [8, 9]. It simulates a practical setting where models must perform online update on every incoming example, without accumulating examples within a task.

Following above definitions, our goal is to learn a classification model $f(x; \theta)$ on the data stream $D$ that can preserve its performance over all the tasks in $D$. At time step $t$ during the training process, the model is updated to minimize a predefined empirical loss $\ell(x_t, y_t; \theta)$ on the newly received example $(x_t, y_t)$, without increasing the loss on the previously visited examples (before $t$). Specifically, let $p_c(x, y; T)$ denotes the distribution of the examples visited until the time step $T$. We look to minimize the expected loss $\mathbb{E}_{p_c(x, y; T)} \ell(x, y; \theta)$ – *i.e.*, retaining performance on tasks encountered before $T$.

### 3.2 Gradient based Memory Editing (GMED)

In online task-free continual learning, examples visited earlier cannot be accessed (revisited) and thus computing the loss over all the visited examples (in $D$) is not possible. To deal with this challenge, memory-based CL algorithms store and (continuously) maintain a set of visited examples in a fixed-size memory and use them for replay or regularization in the future training steps [26, 33]. For instance, in Experience Replay (ER) [33] the examples are randomly sampled from the memory for replay; whereas recent works explore more sophisticated replay strategies, such as Maximally Interfered Retrieval with experience replay (ER-MIR) [2] where memory examples which interfere the most with the newly visited example are selected for replay. However, these algorithms only train over samples drawn from a small replay memory in their original form. As such, the utility of the stored examples could diminish over time as the model could potentially memorize these examples if the memory examples are refreshed in a slow pace – which is found often the case [5].

We address the above limitation in our proposed approach Gradient based Memory Editing (GMED) by allowing examples stored in the memory to be edited in the continuous input space, illustrated in Figure 1. The editing step is guided by an optimization objective instead of being pre-defined and involves drawing examples from the memory, editing the drawn examples, replaying the edited

examples and at the same time writing the edited examples back to the memory. We now state our optimization objectives of example editing followed by algorithmic details of GMED.

**Editing Objective.** Clearly, the most crucial step involved in GMED is identifying "*how*" should the stored examples in the memory be edited to reduce catastrophic forgetting of early examples. If $(x_m, y_m)$ denotes an example drawn from the memory $M$, the goal is to design a suitable editing function $\phi$ to generate the edited example $\hat{x}_m$ where $\hat{x}_m = \phi(x_m)$. Such "editing" process can also be found in white-box adversarial example construction for robustness literature [13] where the task is to find an "adversarial" example $(x', y)$ that is close to an original example $(x, y)$ but is mis-classified by the model. Typically, adversarial example construction utilizes the gradients obtained from the classifier and edits the examples to move towards a different target class. While, in the context of continual learning, the editing objective should be different. We employ a similar hypothesis as previous works [2, 7, 40] that examples that are likely forgotten by models should be prioritized for replay. Accordingly, we edit examples so that they are more likely to be forgotten in future updates. With this objective of editing in place, we detail the process of incorporating GMED with Experience Replay (ER).

---

**Algorithm 1:** Gradient Memory EDiting with ER (ER+GMED)

1: **Input:** learning rate $\tau$, edit stride $\alpha$, regularization strength $\beta$, decay rate $\gamma$, model parameters $\theta$
2: **Receives**: stream example $(x_D, y_D)$
3: **Initialize**: replay memory $M$
4: **for** $t = 1$ **to** $T$ **do**
5:    //when $\gamma = 1$, $k$ is not required
6:    $(x_m, y_m) \sim M$; $k \leftarrow$ replayed_time$(x_m, y_m)$
7:    $\ell_{\text{before}} \leftarrow \text{loss}(x_m, y_m, \theta_t)$;
8:    $\ell_{\text{stream}} \leftarrow \text{loss}(x_D, y_D, \theta_t)$
9:    //update model with stream examples
10:   $\theta'_t \leftarrow \text{SGD}(\ell_{\text{stream}}, \theta_t, \tau)$
11:   //evaluate forgetting of memory examples
12:   $\ell_{\text{after}} \leftarrow \text{loss}(x_m, y_m, \theta'_t)$
13:   $d \leftarrow \ell_{\text{after}} - \ell_{\text{before}}$
14:   //edit memory examples
15:   $x'_m \leftarrow x_m + \gamma^k \alpha \nabla_x (d - \beta \ell_{\text{before}})$
16:   $\ell = \text{loss}(\{(x'_m, y_m), (x_D, y_D)\}, \theta_t)$
17:   $\theta_{t+1} \leftarrow \text{SGD}(\ell, \theta_t, \tau)$
18:   replace $(x_m, y_m)$ with $(x'_m, y_m)$ in $M$
19:   reservoir_update$(x_D, y_D, M)$
20: **end for**

---

### 3.3 The GMED Algorithm with ER

Algorithm 1 summarizes the process and Figure 1 provides a schematic of the steps involved in GMED. At time step $t$, the model receives a mini-batch of the stream examples $(x_D, y_D)$ from the training stream $D$, and randomly draws a same number of memory examples $(x_m^k, y_m)$ from the memory $M$, which we assume has already been drawn for replay for $k$ times. We first compute the "interference" (*i.e.,* loss increase) on the memory example $(x_m^k, y_m)$ when the model performs one gradient update on parameters with the stream example $(x_D, y_D)$.

$$\theta'_t = \theta_t - \nabla_\theta \ell(x_D, y_D; \theta_t); \tag{1}$$

$$d_{t:t+1}(x_m^k, y_m) = \ell(x_m^k, y_m; \theta'_t) - \ell(x_m^k, y_m; \theta_t), \tag{2}$$

where $\theta_t$ and $\theta'_t$ are model parameters before and after the gradient update respectively. Then, we perform a step of gradient update on $(x_m^k, y_m^k)$ by maximizing its "loss increase" in the next one step of training, while using a regularization term $\ell(x_m, y_m; \theta_t)$ to penalize the loss increase evaluated with the current model checkpoint. The iterative update is written as,

$$x_m^{k+1} \leftarrow x_m^k + \gamma^k \alpha \nabla_x [d_{t:t+1}(x_m^k, y_m) - \beta \ell(x_m^k, y_m; \theta_t)], \tag{3}$$

where the hyper-parameter $\alpha$ controls the overall stride of the edit and is tuned with first three tasks together with the regularization strength $\beta$. $\gamma$ is a decay factor of edit performed on the model. A decay factor $\gamma$ less than 1.0 could effectively prevent $x_m^k$ from drastically deviating from their original state, while $\gamma = 1.0$ indicates no decay. We note that we cannot perform constrained edits that strictly adhere to a distance budget w.r.t original examples, as it requires storing original examples which introduces extra memory overhead.

Following Eq. 3, we perform a gradient update on $x$ to increase its "interference". The algorithm then discards $\theta'_t$, and updates model parameters $\theta_t$ using the edited memory example $(x_m^{k+1}, y_m)$ and the stream example $(x_D, y_D)$, in a similar way to ER.

$$\theta_{t+1} = \theta_t - \nabla_\theta \ell(\{(x_m^{k+1}, y_m), (x_D, y_D)\}; \theta_t). \tag{4}$$

We replace the original examples in the memory with the edited example. In this way, we continuously edit examples stored in the memory alongside training.

### 3.4 Applying GMED with Data Augmentation, MIR and GEM

Since the process to edit the original examples in GMED is modular, we can integrate GMED with a range of existing memory-based CL algorithms. In addition to ER, we also explore ER with data augmentation ($ER_{aug}$) [5], ER-MIR [2] and GEM [26] in our experiments.

$ER_{aug}$ applies standard data augmentations (*e.g.*, random cropping, horizontal flipping, denoted as $\mathcal{T}$) to examples $(x_m, y_m)$ drawn from the memory which are replayed at each time step. In $ER_{aug}$+GMED, we edit original example $x_m$ and replay both the augmented example $\mathcal{T}(x_m)$ and the edited example $\hat{x}_m$. To keep the number of replayed examples the same, for $ER_{aug}$ method, we also replay both the edited example $\mathcal{T}(x_m)$ and the original example $x_m$. Finally, we write the edited example $\hat{x}_m$ to the memory.

For integration with ER-MIR (denoted henceforth as MIR for brevity), recall that MIR retrieves and then replays the most "interfering" examples in the memory at each time-step. Thus, making GMED edits on the MIR-retrieved examples may induce a loop of further increasing "interference" of the examples that are already the most "interfering" ones. Instead, in our MIR+GMED implementation, we edit a mini-batch of examples randomly sampled from the memory — a process that is *independent* of the MIR replay operation[2]. This random sampling may help prevent GMED editing from intensifying potential biases created from the MIR retrieval process (*e.g.*, retrieved examples are edited and thus become more interfered). Similarly, For GEM+GMED, we also apply GMED to edit a mini-batch of randomly sampled examples.

Details of the integrated algorithms (*i.e.*, $ER_{aug}$+GMED, MIR+GMED, and GEM+GMED) can be found in Algorithms 2, 3 and 4 in Appendix B. We leave more sophisticated integration of GMED to existing CL algorithms *e.g.* by optimizing the retrieval and editing operations jointly to future work.

## 4 Experiments

Our experiments address the following research questions: (i) what are the gains obtained by integrating GMED with existing memory-based CL algorithms and how these gains compare across datasets, methods and memory sizes? (ii) how useful are GMED edits in alleviating catastrophic forgetting, and what part of it can be attributed to the design of the editing objective function? (iii) what role do the various components and parameters in the GMED play and how they affect the performance. In the rest of this section, we first briefly describe the datasets (Sec. 4.1) and the compared baselines (Sec. 4.2). We then detail our main results comparing across memory-based algorithms (Sec. 4.3), validate the effectiveness of GMED-edits and the editing objective in mitigating catastrophic forgetting (Sec. 4.4), followed by ablative study over its components (Sec. 4.5).

### 4.1 Datasets

We use six public CL datasets in our experiments. **Split / Permuted / Rotated MNIST** are constructed from the MNIST [21] dataset which contains images of handwritten digits. Split MNIST [12] creates 5 disjoint subsets based on the class labels and considers each subset as a separate task. The goal then is to classify over all 10 digits when the training ends. Permuted MNIST [12] consists of 10 tasks, where for a particular task a random permutation in the pixel space is chosen and applied to all images within that task. The model then has to classify over the 10 digits without knowing which random permutation was applied. Rotated MNIST [26] rotates every sample in MNIST by a fixed angle between 0 to 180. Similar to the previous datasets, the goal is to classify over 10 digits without any knowledge of the angle of rotation. For all MNIST experiments, each task consists of 1,000 training examples following [2]. We also employ **Split CIFAR-10 and Split CIFAR-100**, which comprise of 5 and 20 disjoint subsets respectively based on their class labels. The model then classifies over the space of all class labels. Similarly, **Split mini-ImageNet** [2] splits the mini-ImageNet [10, 42] dataset into 20 disjoint subsets based on their labels. The models classify over all 100 classes.

Following the taxonomy of [41], the Split MNIST, Split CIFAR-10, Split CIFAR-100, and Split mini-ImageNet experiments are categorized under class-incremental setup, while Permuted and Rotated MNIST experiments belong to domain-incremental setup. We note that our results are not comparable

---

[2]This also ensures the integrated approach will replay the same number of examples as the baselines, yielding a fair comparison.

*Table 1:* **Mean and standard deviation of final accuracy (%) for non-model-expansion-based approaches on** 6 **datasets.** For Split mini-ImageNet and Split CIFAR-100 datasets, we set the memory size to 10,000 and 5,000 examples; we use 500 for other datasets. $^*$ and $^{**}$ over GMED methods indicate significant improvement over the counterparts without GMED with $p$-values less than 0.1 and 0.05 respectively in single-tailed paired t-tests. We report results in 20 runs for GEM, ER, MIR, and $ER_{aug}$ and their GMED-integrated versions, and 10 runs for others. $\dagger$ over "previous SOTA" results indicates that the best GMED method (**bolded** for each dataset) outperforms the previous SOTA with statistically significant improvement ($p < 0.1$).

| Methods / Datasets | Split MNIST | Permuted MNIST | Rotated MNIST | Split CIFAR-10 | Split CIFAR-100 | Split mini-ImageNet |
|---|---|---|---|---|---|---|
| Fine tuning | $18.80 \pm 0.6$ | $66.34 \pm 2.6$ | $41.24 \pm 1.5$ | $18.49 \pm 0.2$ | $3.06 \pm 0.2$ | $2.84 \pm 0.4$ |
| AGEM [8] | $29.02 \pm 5.3$ | $72.17 \pm 1.5$ | $50.77 \pm 1.9$ | $18.49 \pm 0.6$ | $2.40 \pm 0.2$ | $2.92 \pm 0.3$ |
| GSS-Greedy [4] | $84.16 \pm 2.6$ | $77.43 \pm 1.4$ | $73.66 \pm 1.1$ | $28.02 \pm 1.3$ | $19.53 \pm 1.3$ | $16.19 \pm 0.7$ |
| BGD [44] | $13.54 \pm 5.1$ | $19.38 \pm 3.0$ | $77.94 \pm 0.9$ | $18.23 \pm 0.5$ | $3.11 \pm 0.2$ | $24.71 \pm 0.8$ |
| ER [33] | $81.07 \pm 2.5$ | $78.65 \pm 0.7$ | $76.71 \pm 1.6$ | $33.30 \pm 3.9$ | $20.11 \pm 1.2$ | $25.92 \pm 1.2$ |
| ER + GMED | $82.67^{**} \pm 1.9$ | $78.86 \pm 0.7$ | $77.09^* \pm 1.3$ | $34.84^{**} \pm 2.2$ | $20.93^* \pm 1.6$ | $27.27^{**} \pm 1.8$ |
| MIR [2] | $85.72 \pm 1.2$ | $79.13 \pm 0.7$ | $77.50 \pm 1.6$ | $34.42 \pm 2.4$ | $20.02 \pm 1.7$ | $25.21 \pm 2.2$ |
| MIR + GMED | $\mathbf{86.52^{**} \pm 1.4}$ | $\mathbf{79.25 \pm 0.8}$ | $79.08^{**} \pm 0.8$ | $36.17^* \pm 2.5$ | $\mathbf{21.22^{**} \pm 1.0}$ | $26.50^{**} \pm 1.3$ |
| $ER_{aug}$ [5] | $80.14 \pm 3.2$ | $78.11 \pm 0.7$ | $80.04 \pm 1.3$ | $46.29 \pm 2.7$ | $18.32 \pm 1.9$ | $30.77 \pm 2.2$ |
| $ER_{aug}$ + GMED | $82.21^{**} \pm 2.9$ | $78.13 \pm 0.6$ | $\mathbf{80.61^* \pm 1.2}$ | $\mathbf{47.47^* \pm 3.2}$ | $19.60 \pm 1.5$ | $\mathbf{31.81^* \pm 1.3}$ |
| Previous SOTA | $85.72^\dagger \pm 1.2$ | $79.23 \pm 0.7$ | $80.04^\dagger \pm 1.3$ | $46.29^\dagger \pm 2.7$ | $20.02^\dagger \pm 1.7$ | $30.77^\dagger \pm 2.2$ |
| iid online | $85.99 \pm 0.3$ | $73.58 \pm 1.5$ | $81.30 \pm 1.3$ | $62.23 \pm 1.5$ | $18.13 \pm 0.8$ | $17.53 \pm 1.6$ |
| iid offline (upper bound) | $93.87 \pm 0.5$ | $87.40 \pm 1.1$ | $91.38 \pm 0.7$ | $76.36 \pm 0.9$ | $42.00 \pm 0.9$ | $37.46 \pm 1.3$ |

to works that employ a different setup over the same dataset (*e.g.*, results on domain-incremental Split CIFAR-100 are not comparable).

## 4.2   Compared Methods

We compare against several task-free memory based continual learning methods namely, Experience Replay (ER) [33], Averaged Gradient Episodic Memory (AGEM) [8], Gradient based Sample Selection (GSS) [4], and Maximally Interfering Retrieval (MIR) [2]. We omit the generative replay method GEN-MIR proposed together in [2] as it underperforms their memory-based counterparts even on simple datasets such as Split MNIST.

We also compare with data augmentation [5] such as random rotations, scaling, and horizontal flipping applied to memory examples drawn for replay in ER, noted as $ER_{aug}$ (except for MNIST datasets). We also include regularization-based, model expansion-based and task-aware approaches, namely Bayesian Graident Descent (BGD) [44], Neural Dirichlet Process Mixture Model (CN-DPM) [22], Progressive Networks (Prog.NN) [36], Compositional Lifelong Learning [29], Graident Episodic Memory (GEM) [26] and Hindsight Anchor Learning (HAL) [7] respectively.

Finally, we report three baseline models: (i) Fine Tuning, where no continual learning algorithms are used for online updates to model parameters, (ii) iid Online, where we randomly shuffle the data stream, so that the model visits an i.i.d. stream of examples, and (iii) iid Offline, where multiple passes over the dataset is allowed. Appendix A provides more details on compared methods and their implementation details. We build our proposed GMED approach upon four baselines, noted as ER+GMED, MIR+GMED, GEM+GMED, and $ER_{aug}$+GMED.

**Implementation Details**. We set the size of replay memory as $10K$ for split CIFAR-100 and split mini-ImageNet, and $500$ for all remaining datasets. Following [8], we tune the hyper-parameters $\alpha$ (editing stride) and $\beta$ (regularization strength) with only the first three tasks. While $\gamma$ (decay rate of the editing stride) is a hyper-parameter that may flexibly control the deviation of edited examples from their original states, we find $\gamma=1.0$ (*i.e.*, no decay) leads to better performance in our experiments. Results under different $\gamma$ setups are provided in Appendix C, and in the remaining sections we assume no decay is applied. For model architectures, we mostly follow the setup of [2]: for the three MNIST datasets, we use a MLP classifier with 2 hidden layers with 400 hidden units each. For Split CIFAR-10, Split CIFAR-100 and Split mini-ImageNet datasets, we use a ResNet-18 classifier with three times less feature maps across all layers. See Appendix C for more details.

## 4.3   Performance Across Datasets

We summarize the results obtained by integrating GMED with different CL algorithms. In Table 1, we report the final accuracy and the standard deviation and make the following key observations.

**Effectiveness of Memory Editing**. As can be seen in Table 1, GMED significantly improves performance on 5 datasets when built upon ER and MIR respectively. The improvement of MIR+GMED

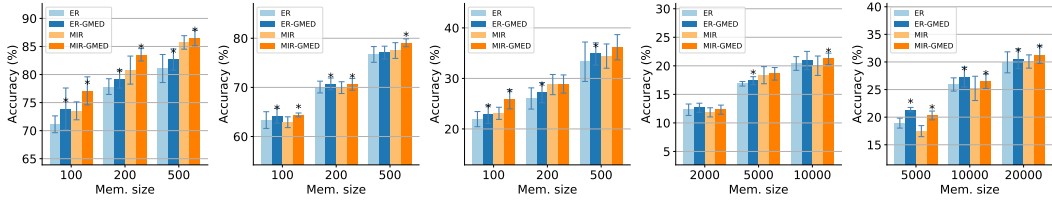

*(a)* Split MNIST    *(b)* Rotated MNIST    *(c)* Split CIFAR-10    *(d)* Split CIFAR-100 *(e)* Split mini-ImgNet

*Figure 2:* **Performance of ER, GMED+ER, MIR, and GMED+MIR with different memory sizes**. For mini-ImageNet dataset, we use memory sizes of $1K$, $5K$, $10K$, and $20K$ examples; for Split CIFAR-100 dataset, we use $1K$, $2K$, $5K$ and $10K$; for other datasets, we use 100, 200, 500, and 1000. $*$ indicates whether the improvement of ER+GMED or MIR+GMED is significant with $p < 0.05$.

corroborates that the optimization in the continuous input spaces of GMED is complementary to sample selection over real examples as in MIR. We also notice significant improvement of $ER_{aug}$ over GMED on 5 datasets. This indicates that the benefits induced by GMED go beyond regularization effects used to mitigate over-fitting.

**Comparison across CL methods.** From Table 1, we find MIR+GMED achieves the best performance on Split MNIST, Permuted MNIST, and Split CIFAR-100 datasets; while on Rotated MNIST, Split CIFAR-10 and Split mini-ImageNet dataset, $ER_{aug}$+GMED achieves the best performance. Performance of the best performing GMED method could significantly improve over previous SOTA on five datasets.

We further compare with an non-memory based CL approach, CN-DPM [22], which employs a generative model, a dynamically expanding classifier, and utilizes a short-term

*Table 2:* **Comparison with model-expansion-based approaches** under the same memory overhead as CN-DPM. The overhead is the size of the replay memory plus the extra model components (e.g. a generator or modules to solve individual tasks), shown in the equivalent number of memory examples (#. Mem). † indicates quoted numbers are taken from the respective papers.

| Method | Split MNIST | | Split CIFAR-10 | | Split CIFAR-100 | |
|---|---|---|---|---|---|---|
| | Acc. | #. Mem | Acc. | #. Mem | Acc. | #. Mem |
| **CN-DPM**† | 93.23 | 2,581 | 45.21 | 6,024 | 20.10 | 21,295 |
| **Prog. NN** | 89.46 | 9,755 | 49.68 | 6,604 | 19.17 | 31,766 |
| **CompCL** | 91.27 | 9,755 | 45.62 | 6,604 | 20.51 | 31,766 |
| **ER** | 92.67 | 2,581 | 62.96 | 6,024 | 21.79 | 21,295 |
| **ER+GMED** | **94.16** | 2,581 | **63.28** | 6,024 | **22.12** | 21,295 |

memory (STM). Following the setup in [16], we set the memory size for GMED so that two methods introduces the same amount of the overhead. Table 2 shows the results of ER, ER+GMED and the reported results of CN-DPM. Interestingly, ER by itself achieves comparable performance to model expansion approaches. ER+GMED further outperforms CN-DPM without any extra memory overhead compared to ER. Similarly, GMED outperforms *task-aware* model expansion approaches such as Prog. NN and the recently proposed compositional model expansion (CompCL) with a smaller memory overhead.

**Performance under Various Memory Sizes.** Figure 2 shows the performance of ER, ER+GMED, MIR, and MIR+GMED under various memory sizes. The improvement on Split MNIST and Split mini-ImageNet are significant with $p < 0.05$ over all memory size setups. On Rotated MNSIT and Split CIFAR-10 the improvements are also mostly significant. The improvement on Split CIFAR-100 is less competitive, probably because the dataset is overly difficult for class-incremental learning, from the accuracy around or less than $20\%$ in all setups.

**Performance on Data Streams with Fuzzy Task Boundaries**. The experiments in Table 1

*Table 3:* **Performance of methods over data streams with fuzzy task boundaries**. In this setup, examples from the next tasks are introduced and gradually dominate the stream when half of the examples from the current task is visited. * indicates whether the improvement is significant ($p < 0.05$)

| Methods / Datasets | Split MNIST | Split CIFAR-10 | Split mini-ImageNet |
|---|---|---|---|
| **Vanilla** | $21.53 \pm 0.1$ | $20.69 \pm 2.4$ | $3.05 \pm 0.6$ |
| **ER** | $79.74 \pm 4.0$ | $37.15 \pm 1.6$ | $26.47 \pm 2.3$ |
| **MIR** | $84.80 \pm 1.9$ | $38.70 \pm 1.7$ | $25.83 \pm 1.5$ |
| $\mathbf{ER}_{aug}$ | $81.30 \pm 2.0$ | $47.97 \pm 3.5$ | $30.75 \pm 1.0$ |
| **ER + GMED** | $82.73^* \pm 2.6$ | $40.57^* \pm 1.7$ | $28.20^* \pm 0.6$ |
| **MIR + GMED** | $\mathbf{86.17 \pm 1.7}$ | $41.22^* \pm 1.1$ | $26.86^* \pm 0.7$ |
| $\mathbf{ER}_{aug}$ **+ GMED** | $82.39^* \pm 3.7$ | $\mathbf{51.38^* \pm 2.2}$ | $\mathbf{31.83 \pm 0.8}$ |

assume a clear task boundary. In Table 3, we report the result of using data-streams with fuzzy task boundaries on four datasets. We leave the complete results in Table 9 in Appendix. In this setup, the probability density of a new task grows linearly starting from the point where 50% of examples of

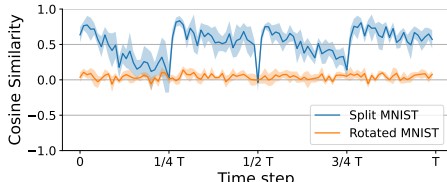

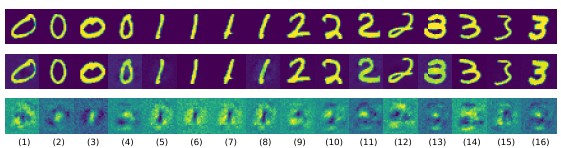

*Figure 3:* Cosine similarity between the optimal editing direction and direction from ER+GMED over training. The averaged similarity is $0.523 \pm 0.014$ and $0.035 \pm 0.009$.

*Figure 4:* Visualization of the edited examples in Split MNIST in the ER-GMED approach. The first two rows show examples before and after editing, and the third row shows the differences.

the current task are visited. In particular, GMED improves performance in three datasets across ER, MIR and $ER_{aug}$.

**Effectiveness of GMED in Task-Aware Setting.** We additionally compare performance of other task-aware approaches (HAL, GEM, GEM+GMED) in Appendix H.

### 4.4 Ablation to Study the Effect of Memory-Editing

We present a set of experiments to validate that the gains obtained through the memory-editing step of GMED are indeed helpful towards alleviating catastrophic forgetting. Further, we show that these gains are distinct from those obtained through random perturbations or simple regularization effects.

**Comparison to Optimal Editing Direction.** At a particular time-step $t$, GMED identifies the edit direction using the stream example $(x_D, y_D)$ and the memory sample $(x_m, y_m)$. To validate whether the edits proposed by GMED are indeed helpful, we compare GMED-edit to an "Optimal Edit" that minimizes the loss increase ("interference") of all early training examples in one future time step.

To compute this Optimal Edit, we note that the total loss increase over all previously encountered examples would be $d_{t:t+1}^{1:t} = \sum_{i=1}^{t} d_{t:t+1}(x_i, y_i) = \sum_{i=1}^{t} [\ell(x_i, y_i; \theta_{t+1}) - \ell(x_i, y_i; \theta_t)]$, where $\theta_{t+1} = \theta_t - \nabla_\theta \ell(x_m, y_m; \theta_t) - \nabla_\theta \ell(x_D, y_D; \theta_t)$ is the model parameters after the training update on $(x_D, y_D)$ and $(x_m, y_m)$. The Optimal Edit direction for the memory example $x_m$ would be the gradient of $d_{t:t+1}^{1:t}$ w.r.t. $x_m$. Computing such optimal edits requires access to early training examples and is not practical in an online continual learning setup; we present it only for ablative purposes.

Figure 3 shows the cosine similarity of update directions (*i.e.*, the gradient of $x_m$) between GMED and the optimal editing strategy over Split MNIST and Rotated MNIST datasets. The averaged similarity over time is $0.523 \pm 0.014$, $0.035 \pm 0.009$ on two datasets, averaged across 10 runs. Recall that random editing has an expectation of zero similarity. On Split MNIST dataset, where the improvement is the most significant, we notice a high similarity between the update directions of GMED and optimal editing. On Rotated MNIST where the improvement is less significant, the similarity is still positive on average but is lower than Split MNIST. It implies GMED-edits are generally consistent with explicitly reducing forgetting, but there is still space to improve. The results also imply the whether the edits of GMED aligns with editing is highly dependent on certain properties of datasets. We further include the classification performance of optimal editing in Appendix L.

**Comparison with alternative Editing Objectives**. While GMED objective editing correlates with that of Optimal Edit, we further validate the choice of the objective function and consider two alternatives to the proposed editing objective (Eq. 3): (i) Random Edit: memory examples are updated in a random direction with a fixed stride; (ii) Adversarial Edit: following the literature of adversarial example construction [13], the edit increases the loss of memory by following the gradient sgn $\nabla_x \ell(x_m, y_m; \theta)$. We report the comparison in Table 4.

We notice ER+Random Edit outperforms ER on split mini-ImageNet, which indicates adding random noise to memory examples helps in regularization. Even so, GMED-edits are as good or

*Table 4:* **Alternative memory editing objectives or simply increasing the number of replayed examples**. Comparison of Random Edit, Adversarial Edit (Adv. Edit) to our proposed objective in GMED. * indicates significant improvement ($p<0.05$) compared to adversarial edit.

| Methods / Datasets | Rotated MNIST | Split CIFAR-10 | Split mini-ImageNet |
|---|---|---|---|
| ER | $76.71 \pm 1.6$ | $33.30 \pm 3.9$ | $25.92 \pm 1.2$ |
| ER + Random Edit | $76.42 \pm 1.3$ | $32.26 \pm 1.9$ | $26.50 \pm 1.3$ |
| ER + Adv. Edit | $76.13 \pm 1.5$ | $31.69 \pm 0.8$ | $26.09 \pm 1.6$ |
| ER + GMED | $\mathbf{77.50 \pm 1.6}$ | $\mathbf{34.84^* \pm 2.2}$ | $\mathbf{27.27^* \pm 1.8}$ |
| MIR+Extra 1 batch | $78.07 \pm 1.1$ | $33.81 \pm 2.3$ | $24.94 \pm 1.5$ |
| MIR+Extra 2 batches | $77.10 \pm 1.4$ | $32.36 \pm 2.8$ | $24.65 \pm 1.5$ |
| MIR | $77.50 \pm 1.6$ | $34.42 \pm 2.4$ | $25.21 \pm 2.2$ |
| MIR + Random Edit | $77.19 \pm 1.0$ | $35.39 \pm 3.0$ | $24.86 \pm 0.7$ |
| MIR + Adv. Edit | $78.06 \pm 1.5$ | $35.79 \pm 0.4$ | $25.48 \pm 1.3$ |
| MIR + GMED | $\mathbf{79.08^* \pm 0.8}$ | $\mathbf{36.17 \pm 2.5}$ | $\mathbf{26.29^* \pm 1.2}$ |

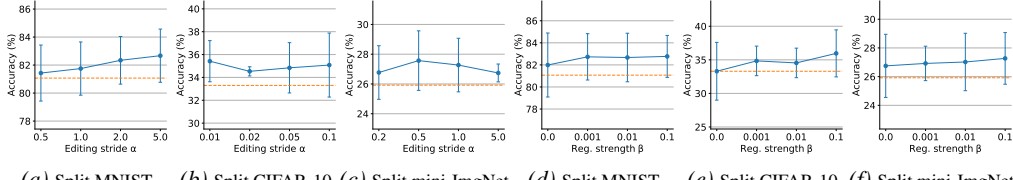

*(a)* Split MNIST    *(b)* Split CIFAR-10   *(c)* Split mini-ImgNet    *(d)* Split MNIST    *(e)* Split CIFAR-10   *(f)* Split mini-ImgNet

*Figure 5:* **Parameter sensitivity analysis** of the editing stride $\alpha$ (a,b,c) and the regularization strength $\beta$ (d,e,f) for ER+GMED across various datasets. The dashed horizontal line indicates the performance of ER.

outperforms both random and adversarial edits across multiple datasets and models. This validates our choice of the editing objective.

**Comparison to Increasing the Number of Replayed Examples.** Recall that in MIR-GMED, we sample two independent subset of memory examples to perform editing and replay (Sec. 3.4). For a fairer comparison, we replay one (or more) extra subset of examples. Table 4 shows the performance does not improve as replaying more examples. We hypothesize that the performance is bottle-necked by the size of the memory and not the number of examples replayed.

**Case study on Edited Memory Examples.** In Figure 4, we show visualizations of editing of memory examples. The examples are drawn from first two task (0/1, 2/3) in the Split MNIST dataset using ER+GMED. The first and second row show the original and edited examples, noted as $x_{\text{before}}$ and $x_{\text{after}}$ respectively. The third row shows the difference between two $\Delta x = x_{\text{after}} - x_{\text{before}}$. While there is no significant visual differences between original and edited examples, in the difference $\Delta x$, we find examples with exaggerated contours (*e.g.* examples (1) and (12)) and blurring (*e.g.* examples (2), (3), (5), and (6)). Intuitively, ambiguous examples are exaggerated while typical examples are blurred. Our visualizations supports this intuition: examples (1) and (12) are not typically written digits, while examples (2), (3), (5), and (6) are typical. Appendix D provides a t-SNE [27] plot of the edit vector.

## 4.5 Analysis on the GMED Framework

Here, we analyze and ablate the various components used in the GMED framework and their effect on the final performance.

**Hyper-parameter Sensitivity**. In Figure 5 we plot the sensitivity of the performance of GMED with respect to two hyper-parameters: the editing stride $\alpha$ and the regularization strength $\beta$ (Eq. 3). Clearly, ER+GMED outperforms ER over a broad range of $\alpha$ and $\beta$ setups. Furthermore, in Figure 5 (d,e,f), better performance for non-zero $\beta$ confirms the benefit of the regularization term. Recall that in our main experiments we tune the hyper-parameters $\alpha$ and $\beta$ with only first three tasks; we note the chosen hyperparameters improve the performance despite they are not always optimal ones.

**Computational Efficiency.** We analyze the *additional* forward and backward computation required by ER+GMED and MIR. Compared to ER, ER+GMED adds 3 forward and 1 backward passes to estimate loss increase, and 1 backward pass to update the example. In comparison, MIR adds 3 forward and 1 backward passes with 2 of the forward passes are over a larger set of retrieval candidates. In our experiments, we found GMED has similar training time cost as MIR. In Appendix B, we report the wall-clock time, and observe the run-time of ER+GMED is 1.5 times of ER.

**Increasing Steps of Editing.** For experiments in Table 1, we performed one editing step over the sampled memory example $x_m$ at time step $t$. In general, we can increase the number of editing steps. The direction for the edit is computed at each step, which makes the approach different from increasing the editing stride ($\alpha$). Table 5 indicates that 3-step and 5-step edits in general don't

*Table 5:* **Editing examples for multiple gradient steps in ER+GMED.** We tune the editing stride ($\alpha$) using the first three tasks as described in Sec. 3.3.

| Methods / Datasets | Split MNIST | Rotated MNIST | Split CIFAR-10 | Split mini-ImageNet |
|---|---|---|---|---|
| **1-step Edit** | $82.21 \pm 2.9$ | $77.50 \pm 1.6$ | $34.84 \pm 2.2$ | $27.27 \pm 1.8$ |
| **3-step Edit** | $82.55 \pm 1.9$ | $77.37 \pm 1.7$ | $34.93 \pm 1.4$ | $\mathbf{27.36 \pm 1.7}$ |
| **5-step Edit** | $\mathbf{83.11 \pm 1.9}$ | $\mathbf{77.53 \pm 1.6}$ | $\mathbf{36.82 \pm 1.8}$ | $26.36 \pm 2.0$ |

lead to significant improvement in performance while incurring additional computational cost. As such, we use only 1-edit step across all our experiments.

## 5 Conclusion

In this paper, we propose Gradient based Memory Editing (GMED), a modular framework for memory-based task-free continual learning where examples stored in the memory can be edited. Importantly, memory examples edited by GMED are encouraged to remain in-distribution but yield increased loss in the upcoming model updates, and thus are more effective at alleviating catastrophic forgetting. We find that combining GMED with existing memory-based CL approaches leads to consistent improvements across multiple benchmark datasets, only inducing a small computation overhead. Finally, we perform a thorough ablative study to validate that the gains obtained by GMED can indeed be attributed to its editing operation and careful choice of editing objective.

## Acknowledgements

This research is supported in part by the Office of the Director of National Intelligence (ODNI), Intelligence Advanced Research Projects Activity (IARPA), via Contract No. 2019-19051600007, the DARPA MCS program under Contract No. N660011924033, the Defense Advanced Research Projects Agency with award W911NF-19-20271, NSF IIS 2048211, NSF SMA 1829268, and gift awards from Google, Amazon, JP Morgan and Sony. We would like to thank all the collaborators in USC INK research lab for their constructive feedback on the work.

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
