# OpenReview forum: "Gradient-based Editing of Memory Examples for Online Task-free Continual Learning"
_NeurIPS.cc/2021/Conference — NeurIPS 2021 Poster_

### Official Review · Reviewer_CT4C · 2021-07-13

**Rating:** 6
**Confidence:** 4

**Summary:**

In this paper, the authors propose a method called Gradient based Memory EDiting (GMED) for task-free continual learning. The motivation is to address one drawback of existing experience replay methods for continual learning, that is, the utility of the stored examples may diminish over time during training. The idea is to update the example in the memory to maximize its interference. The proposed method can be combined with existing continual learning methods to further boost the performance. The authors conduct extensive experiments and ablation studies to show the benefits of the approach.

**Limitations And Societal Impact:**

Yes

**Main Review:**


Originality: The idea of modifying examples in the memory for continual learning is novel and interesting. Also, the proposed method is well motivated and simple to implement.


Quality: The paper is technically sound and the authors include extensive ablation studies to show the benefit of the proposed GMED-edits.


Clarity: The paper is generally well-written and the method is clearly presented.


Significance: The improvements are not significant but meaningful. And the proposed method can be easily integrated with exising continual learning methods which is another benefit.


Questions:

1. It would be interesting to see the predictions of the model on the edited examples, i.e., would the edits change the model predictions?

2. It seems that how the examples are selected to store in the memory is another factor which can affect the performance of the method. Several strategies exist: i. Random Sampling, ii. Reservoir Sampling iii. GSS [1]. It would be interesting to see how the ways of storing examples would change the performance of the method.




[1] Gradient based sample selection for online continual learning. 	Neurips 2019


**Time Spent Reviewing:**

3 hours

---

> ### Author Response · Authors · 2021-08-10
> **Response to Reviewer CT4C**
>
> We thank the reviewer for the positive review and thoughtful comments!
>
> ### Q1: It would be interesting to see the predictions of the model on the edited examples, i.e., would the edits change the model predictions?
> Thank you for the great suggestion! We agree with the reviewer and will include the model predictions in our final version of the paper.
>
> ### Q2: It would be interesting to see how the ways of storing examples would change the performance of the method
> We thank the reviewer for the suggestion. We note that the storage mechanism is orthogonal to our proposed work. In our final version, we will combine GMED with other strategies to store examples (e.g., GSS[1]) and analyze the interaction between the two.
>
> [1] Aljundi et al. Gradient based sample selection for online continual learning. 2019

---

> > ### Author Response · Authors · 2021-08-12
> > **Response to Reviewer CT4C (Cont.)**
> >
> > ### [Updated] Q1: It would be interesting to see the predictions of the model on the edited examples, i.e., would the edits change the model predictions?
> >
> > As suggested by the reviewer, we have computed the percentage of changed predictions after example edits. Specifically, when the training completes, we classify examples stored in the memory that have experienced editing at least once and also classify the corresponding original examples. We then compute “prediction change rate” as the portion of changed predictions over all edited examples stored in the memory. We summarize the results in the table below.
> >
> > Dataset | Split MNIST | Rotated MNIST | Split CIFAR-10 | mini-ImageNet
> > ---------- | ----------------- | ------------------- | ------------------ | ---------------
> > Prediction Change Rate |   10.2%        |         2.4%        |           5.1%          |    5.5%
> >
> >
> > We notice that the prediction change rate is positively correlated with performance improvement of GMED over four datasets. We will perform additional human evaluation over the edited examples in our final version to better understand the effect of editing,  and the relationship between the performance and prediction change rate.

---

> > > ### Author Response · Authors · 2021-08-27
> > > **Response to Reviewer CT4C (Cont.)**
> > >
> > > ### [Updated] Q2: It would be interesting to see how the ways of storing examples would change the performance of the method.
> > >
> > > We thank the reviewer for their suggestion on experimenting with different storing mechanisms. We provide quantitative comparisons on two datasets (Split-MNIST and Rotated MNIST) using the GSS-Greedy approach [Ref1] which has the best reported results.
> > >
> > > \# |   Split MNIST | Rotated MNIST
> > > --- | ----- | ----
> > > GSS-Greedy | 83.70±1.0 | 73.80±1.6
> > > GSS-Greedy+GMED | 84.62±1.2 | 74.47±1.3
> > > p-value  | 0.016 | 0.18
> > >
> > > Our results suggest that naively using GSS with GMED leads to improvement on Split-MNIST but not significant on Rotated-MNIST.

---

> ### Author Response · Authors · 2021-08-16
> **Overview of Responses to Reviewer CT4C**
>
> Hi Reviewer CT4C
>
> Thank you again for your detailed review. We are glad that you found our idea to be interesting, novel, well-motivated, and simple to implement. We are also encouraged to know that you found our paper technically sound and supported with extensive experiments. For convenience, please find below an overview of the points covered in the rebuttal:
>
> * Experiments with model predictions for the edited examples.
> * Discussion of ways of storing which would be included in the final version.
> * In general response, we provide a discussion about the improvements by GMED.
>
> We hope that our rebuttal adequately addresses your concerns and questions. Please feel free to provide any additional comments, and we would be happy to address them in the rebuttal as well as in the final version of our paper.
>
> Sincerely,
>
> Paper1846 Authors

---

> > ### Comment · Reviewer_CT4C · 2021-08-27
> > **Response**
> >
> > Thanks for the clarification and more experimental results. It would be nice if the authors can include the results in the final draft. The idea is interesting but the application is somewhat limited to memory-based lifelong learning so I will keep my score of 6.

---

> > > ### Author Response · Authors · 2021-09-01
> > > **Thanks for the replies**
> > >
> > > Yes, we would carefully revise the final draft and include the reported results. We would like to emphasize that memory-based continual learning algorithms achieve state-of-art results across a number of benchmarks (L266) and are broadly applicable in various setups. This is more pronounced in the context of task-free continual learning which is challenging and has high practical applications. As such, we believe GMED which can be plugged into other memory-based CL to improve performance are impactful in scope.

---

### Official Review · Reviewer_ggFa · 2021-07-14

**Rating:** 7
**Confidence:** 4

**Summary:**

This paper proposes a replay-based method (GMED) for tackling the challenging and important problem of task-free continual learning (CL), which is concerned with mitigating catastrophic forgetting when learning from a non-i.i.d. stream of data without knowledge of task boundaries. The key idea of the method is to *edit* samples from the replay buffer during training in a way that maximises the interference that would be caused to the performance on these examples by a gradient update on the minibatch from the current task. The method is motivated by previous work that has shown that prioritising replay of samples that would be most interfered with mitigates catastrophic forgetting (e.g.[1]).

Experiments are run demonstrating that GMED yields a statistically significant performance increase on a number of standard image recognition CL tasks (with discrete task boundaries and also with fuzzy task boundaries, which is more challenging and less common) when combined with a handful of existing replay-based methods ((namely experience replay (ER), ER with augmentations and Maximally Interfered Retrieval (MIR)). Further experiments are run to show that the method is robust to different memory sizes and to a range of values for the hyperparameters that the method uses to balance terms in the editing update. Additionally, experiments are also run to show that the type of editing used by GMED results in a better performance than random edits or adversarial edits to replay samples.

[1] Aljundi, Rahaf, et al. "Online Continual Learning with Maximal Interfered Retrieval." Advances in Neural Information Processing Systems 32 (2019): 11849-11860.

**Limitations And Societal Impact:**

Yes.

**Main Review:**

POST-REBUTTAL update:
I am increasing my score to a 7, after discussion with the authors has provided me with a better intuition of the motivation for editing examples, eg with reference to adversarial example construction, and the assurance from the authors that the paper will be updated accordingly to elaborate on the motivation behind the method.

------------------------
This paper tackles a particularly challenging version of the CL problem, *task-free* continual learning, which is of great interest to the CL community as it moves closer to a real-world setting of streaming data with no task boundaries. One of the main positives of the paper is the extensive set of experiments demonstrating the benefit of the method over a range of tasks, with comparisons to several other methods, along with a good range of ablations and sensitivity analyses. The main drawbacks are that (i) the method has not got a strong theoretical justification and thus it’s not obvious why it works, and (ii) while many of the performance improvements are statistically significant, they are not that large in magnitude and most fall within a standard deviation of the baseline performance. Overall, I recommend for acceptance on the margin as I think the paper demonstrates that editing replay memory might be a promising direction for (a particularly challenging version of) continual learning and could inspire the community to pursue it.

Comments:
- Lacking theoretical grounding. The loss function for the edit updates does not have a clear link to the main objective of mitigating forgetting on previous data. It’s not intuitive that modifying datapoints as GMED does should help to prevent forgetting since it moves them away from the original distribution of data - eventually one might imagine that this would increase forgetting if they move far enough away from the original datapoints. Encouragingly, the paper shows experiments that implement multiple edits (up to 5) per example, which in most cases marginally improves performance (at a computational cost), so perhaps this is not an issue. There is also a visualisation of some examples of edits, showing that there is some blurring or exaggeration of contours, but it’s hard to intuit why these would be helpful. Ideally, some more work would be done to understand why the model works, otherwise it is difficult to build on.
- It was useful to run experiments that increased the number of replayed examples with MIR - this shows to some extent that the benefit of GMED is not simply due to a shift in the balance of replay updates to current task updates during training.
- Having to tune the hyperparameters over the first three tasks is a limitation, though it has precedence in at least one paper as pointed out by the authors (A-GEM).
- It was an interesting idea to correlate GMED’s edits with those of the optimal editing direction (that minimises forgetting on previous tasks by “cheating” and calculating with past data). Is there any intuition for why GMED aligns with the optimal editing direction better for split mnist vs rotated mnist? It would be interesting to see how the optimal editing direction version performs during training, to gauge what the upper bound of memory-editing is.
- While the results show that combining GMED with other algorithms (namely experience replay (ER), ER with augmentations and Maximally Interfered retrieval) yields a statistically significant improvement in performance on numerous tasks, there are a few caveats: the absolute improvement is marginal (on the order of a percentage point), in each case the improvement falls well within 1 s.d. Of the baseline performance, and some of the improvements are only significant with p=0.1, which is an atypically large p-value.

Typos/grammar:
- Line 125. “Found often the case” -> “found often to be the case”.
- Line 154 “an” -> “a”
- Line 181-182. Do you mean to say that you replay the original example and the *augmented* example, rather than the *edited* example, since you are talking about ER_aug, which does not have example editing?
- Line 235 and 236. “Graident” -> “Gradient”
- Line 290. “MNSIT” -> “MNIST”


**Time Spent Reviewing:**

8

---

> ### Author Response · Authors · 2021-08-10
> **Response to Reviewer ggFa**
>
> We thank the reviewer for the positive review and thoughtful comments!
>
> ### Q1: Lack of theoretical grounding
> Thank you for the suggestion. We agree with the reviewers’ comment that the current version of our work doesn’t provide a strong theoretical analysis regarding why GMED can improve the base model. As a first step, we experimented with optimal editing gradients (Section 4.4) which require storing all the samples seen during training. We observe that GMED edits are in general positively correlated with the optimal edits. Moreover, we observe that a higher similarity of GMED edits results in better improvement in performance: ER+GMED improves over ER with p<0.01 on split MNIST and p<0.1 on Rotated MNIST (see Table 1). We hypothesize that with additional assumptions on the data and the data-stream, one could derive a bound for the gap between GMED edits and optimal edits and can further quantify the performance gap between GMED and the oracle method (i.e., based on optimal editing). We leave such detailed analysis to future work.
>
> ### Q2: Having to tune the hyperparameters over the first three tasks is a limitation
> We agree with the reviewer that GMED introduces additional tunable hyperparameters, as compared with its base model. However, in Figure 5, we show that GMED brings performance improvement in a modest range of hyperparameter setup. We will conduct a more systematic analysis on the sensitivity of hyper-parameters introduced by GMED in our final version.
>
> ### Q3: Why GMED aligns with the optimal editing direction better on Split MNIST
> Thank you for raising this insightful question. One hypothesis we had for worse performance in Rotated MNIST is that the average of rotated examples would look widely different when considering all previous examples compared to viewing a small mini-batch of edited examples. We will include more thorough analysis on this observation in our final version.
>
> ### Q4: It would be interesting to see how the optimal editing direction version performs during training, to gauge what the upper bound of memory-editing is.
>
> We appreciate the insightful question from the reviewer. Here is a summary of the performance comparison between ER, ER+GMED and ER+Optimal Edits on Split MNIST and Rotated MNIST.
>
> Method/Dataset | Split MNIST | Rotated MNIST | Split CIFAR-10
> --------------------- | ---------------- | --------------------  | ---------------------
> ER                     |   80.14±3.2  |    76.71±1.6       |  33.30±3.9
> ER+GMED        |   82.67±1.9  |    77.09±1.3       |  34.84±2.2
> ER+Optimal      |    83.40±2.6  |    77.73±1.3      |  35.04±2.6
>
> We notice ER+optimal edit outperforms ER+GMED. Computing optimal edits is rather expensive as it requires computing loss changes over a large sample of previously seen examples in every editing step; we will include complete results over other datasets in the final version.
>
> Finally, we note that our ER+Optimal is not necessarily an upper-bound of memory editing, as it computes optimal edit for only the coming *one* training step over the data stream. Because the edited examples may in turn affect training dynamics in future training steps, it is hard to derive an exact upper bound of memory editing.
>
> ### Q5: The absolute improvement of GMED is marginal.
> Thank you for pointing this out. We refer the reviewer to our response under General Response to reviewers. We will include more discussion regarding this in our final version.
>
> ### Q6: Line 181-182. Do you mean to say that you replay the original example and the augmented example in L181-182 instead of the “edited example”?
> Yes, the “edited example” should have been the “augmented example” in L181-182. We thank the reviewer for pointing out this typo (and also other typos).

---

> > ### Comment · Reviewer_ggFa · 2021-08-13
> > **Thanks for replies - still confused about intuition for method**
> >
> > - Thank you for your replies.
> > - With regards to the theoretical grounding of the method, I am still not sure what the intuition is for why editing the data in the replay buffer helps at all. In other words, why is it intuitive to start experimenting with the optimal edits in the first place, given that this changes the underlying data distribution?
> > - Thank you for showing the results with the optimal edit - this is helpful.

---

> > > ### Author Response · Authors · 2021-08-16
> > > **Response to Reviewer ggFa**
> > >
> > > We appreciate the reviewer for the follow-up comment!
> > >
> > > ### Q: What is the intuition of experimenting with example editing / optimal editing in the first place, given that it changes the underlying data distribution?
> > >
> > > GMED is inspired by prior methods such as MIR [Ref1] and empirical study [Ref2] showing that “interfered examples”  (i.e. memory examples that suffer from increased loss due to model updates) are helpful when replayed (Line 41). The intuition here is that the examples which lead to increased loss are likely to be forgotten by the more recent model (updated on new examples) and therefore replaying such examples is effective against forgetting. Following this intuition, MIR [Ref1] proposes to prioritize replaying of most interfering examples stored in the memory. However, we believe one limitation of MIR is that its performance gain will gradually saturate after rounds of replays on the stored examples ---i.e., these examples may lead to marginal increased loss --- and thus none of the examples in the memory are interfering anymore. This motivates our work to perform editing over the stored examples (in addition to just retrieving them) as in MIR+GMED. In contrast to works like MIR which rank existing memory samples and choose from them, we directly edit randomly selected memory samples.
> > >
> > > Our “editing” process is inspired from adversarial example construction for robustness (Line 89, also included as a baseline in Table 4) where the task is to find an “adversarial” example $(x^\prime, y)$ that is close to an original example $(x,y)$ but is mis-classified by the model. Typically, adversarial methods (white-box methods) utilize the gradients obtained from the classifier and edit the sample to move towards a different target class. In general, adversarial construction methods would lead to a change in underlying data-distribution but the use of a distance budget (via L2-Norm or L-Inf norm) with respect to original sample prevents large deviation.
> > >
> > > In a similar vein, we utilize the gradient updates but edit the samples towards the direction of increased loss thereby making them more effective against alleviating forgetting. Unlike adversarial construction, GMED doesn’t store the original example and as a result one cannot directly use a distance budget w.r.t original sample. Instead, GMED uses regularization terms and small editing strides to heuristically constrain the distance between edited and original examples $||x^\prime-x||_2^2$. While the underlying distribution changes, the regularization term prevents large deviations from the original sample. We anticipate better regularization techniques may push the performance towards the reported “optimal” editing performance which can be a meaningful future work.
> > >
> > > We will make the discussion on intuition of GMED more detailed in our final version.
> > >
> > >
> > > [Ref1] Aljundi et al.  Online continual learning with maximally interfered retrieval, NeurIPS 2019
> > >
> > > [Ref2] Toneva et al. An empirical study of example forgetting during deep neural network learning, ICLR 2019

---

> > > > ### Comment · Reviewer_ggFa · 2021-08-17
> > > > **Response to authors**
> > > >
> > > > Thanks for this, it has improved my understanding of the intuition of the method

---

> > > > > ### Author Response · Authors · 2021-08-17
> > > > > **Response to Reviewer ggFa**
> > > > >
> > > > > Thank you very much for your insightful questions, Reviewer ggFa! We appreciate this discussion and are glad that you find it helpful. We hope that you can reconsider the evaluation of our work. Please do let us know if you have follow-up questions.

---

> ### Author Response · Authors · 2021-08-16
> **Overview of Responses to Reviewer ggFa**
>
> Hi Reviewer ggFa
>
> Thank you again for your very detailed review. We are glad that you found our set of experiments extensive and that editing replay memory is a promising direction for future work. For convenience, please find below an overview of the points covered in the rebuttal:
>
> * Revisiting optimal edits experiments, which could be helpful for future work in theoretically investigating the effects of memory editing.
> * Clarifying that GMED works over a modest range of Hyper-parameters.
> * Discussion as to why optimal edits better align for Split-MNIST.
> * Experiments with optimal edits to obtain upper-bounds of editing operation.
> * In general response, we provide a discussion about the improvements by GMED.
> * Clarifying a typo.
> * Clarifying the intuitions behind using optimal edits.
>
> We hope that our rebuttal adequately addresses your concerns and questions. Please feel free to provide any additional comments, and we would be happy to address them in the rebuttal as well as in the final version of our paper.
>
> Sincerely,
>
> Paper1846 Authors

---

### Official Review · Reviewer_vw9x · 2021-07-17

**Rating:** 6
**Confidence:** 3

**Summary:**

This paper introduces the Gradient based Memory Editing (GMED) framework for task-free online continual learning that edits stored examples using gradient updates. GMED can be easily applied with other memory-based continual learning algorithms to improve their performance. The paper conducts several experiments to validate their method.

**Limitations And Societal Impact:**

No potential negative societal impact.

**Main Review:**

The GMED framework proposed in this paper is interesting and novel. It helps improve the quality and flexibility of the memory/coreset in continual learning without the need to train a complex generative model as in other replay-based continual learning methods. The paper is well-written and the experiments are extensive.

The main drawback, in my view, is that the accuracy improvements of GMED in the experiments seem modest compared to the versions that don't use GMED. For example, in Table 1, there are several cases where the p-value is greater than 0.05, which is not a significant improvement.

Below are some other comments:
1. In Algorithm 1, do we need to know the number of replays k for each example in the memory?
2. In this algorithm, would it be better to edit all examples in the memory instead of only one example x_m? How about editing a mini-batch of examples?
3. In Section 4.4, what are the accuracies when using the optimal edits? Maybe we can add them into Table 4 for a comparison.

**Time Spent Reviewing:**

3

---

> ### Author Response · Authors · 2021-08-10
> **Response to Reviewer vw9x**
>
> We thank the reviewer for the positive review and thoughtful comments!
>
> ### Q1: In Algorithm 1, is the number of replays $k$ required?
> We apologize for the confusion. In short, No, we don’t require to keep track of the number of replays $k$. The number of replays $k$ is used to compute the decay of the editing stride ($\gamma^k$ in Eq. 3). We discussed the effect of $\gamma$ in Appendix C and found the effect was very minor. Therefore, in our main experiments, we set $\gamma$ as 1 (L248), i.e., no decay, in which case $k$ is no longer required.  We will revise the presentation in Algorithm 1 to improve the clarity.
>
> ### Q2: In this algorithm, would it be better to edit all examples in the memory instead of only one example x_m? How about editing a mini-batch of examples?
> For simplicity of notations, in the paper we used (x, y) to denote both “a single example” and “a mini-batch of examples” (L100). In our experiments, editing was performed at the level of mini-batches, where the size of mini-batches is the same as the training batch size set to 10 (Appendix B, L644). We will revise the writing to make this detail more clear in the final version.
>
> We did not edit all examples in the memory in consideration of the computational cost. We further experimented with editing 3, 5, 10, 50 extra examples (in addition to the original 10 examples) per training step in ER+GMED. Results are summarized in the table below:
>
>
> Method/Dataset | Split MNIST | Rotated MNIST | Split CIFAR-10 | Split mini-ImageNet  |
> -------------------------------------- | ---------------- | ---------------------| ---------------------| --------------------------- |
> ER+GMED (10 examples)     |   82.67±1.9  |    77.09±1.3   |    34.84±2.2  |   27.27±1.8   |
> \+ 3 examples  			|   83.00±2.3  |   77.01±1.6    |    34.87±2.9  |  27.36±1.9   |
> \+ 5 examples   		|   82.78±2.3  |    76.98±1.6   |    35.31±2.2  |   28.00±2.0   |
> \+ 10 examples   		|   82.21±2.2  |    76.51±1.4   |    34.86±2.8  |   27.17±1.8  |
> \+ 50 examples                        |   81.42±2.6  |    76.50±1.3   |    31.79±2.6  |   27.10±1.9
>
>
> We noticed improvement on Split MNIST, Split CIFAR-10, Split mini-ImageNet by slightly increasing the number of edited examples. However, the performance drops when the number of edited examples becomes further larger. We hypothesize that GMED’s improvement does not always increase as the number of editing examples increases. We will include more thorough analysis and discussion about this issue in our final version.
>
> ### Q3: What are the accuracies when using the optimal edits?
> We appreciate the question from the reviewer. Here is a summary of the performance comparison between ER, ER+GMED and ER+Optimal Edits on Split MNIST and Rotated MNIST.
>
>
> Method/Dataset | Split MNIST | Rotated MNIST | Split CIFAR-10
> --------------------- | ---------------- | --------------------  | ---------------------
> ER                     |   80.14±3.2  |    76.71±1.6       |  33.30±3.9
> ER+GMED        |   82.67±1.9  |    77.09±1.3       |  34.84±2.2
> ER+Optimal      |    83.40±2.6  |    77.73±1.3      |  35.04±2.6
>
>
> We notice ER+optimal edit outperforms ER+GMED over all three datasets shown above. We also notice that ER-GMED has close performance as ER-Optimal Edit on RotatedMNIST and Split CIFAR-10. We would like to note that computing optimal edits is quite expensive as it requires computing loss changes over a large set of previously seen examples in every editing step. We will include complete performance comparison over other datasets in the final version.
>
>
> ### Q4: The accuracy improvements of GMED in the experiments seem modest compared to the versions that don't use GMED.
> Thank you for raising this concern. We refer the reviewer to our response under General Response to reviewers. We will include more discussion regarding this in our final version.

---

> > ### Comment · Reviewer_vw9x · 2021-09-03
> > **Thanks.**
> >
> > Thanks for your response and the additional experiments. They have clarified the questions that I had.

---

> > > ### Author Response · Authors · 2021-09-03
> > > **Thank you!**
> > >
> > > Thank you very much for your insightful questions, Reviewer vw9x! We appreciate this discussion and are glad that you find it helpful. We hope that you can reconsider the evaluation of our work. Please do let us know if you have follow-up questions.

---

> ### Author Response · Authors · 2021-08-16
> **Overview of Responses to Reviewer vw9x**
>
> Hi Reviewer vw9x
>
> Thank you again for your detailed review. We are glad that you found our approach to be interesting, novel and our experiments extensive. For convenience, please find below an overview of the points covered in the rebuttal:
> * Clarification about storing the number of replays $k$.
> * Experiments with varying sizes of mini-batch for editing.
> * Results using ER+Optimal edits.
> * In general response, a detailed discussion of improvements by GMED.
>
> We hope that our rebuttal adequately addresses your concerns and questions. Please feel free to provide any additional comments, and we would be happy to address them in the rebuttal as well as in the final version of our paper.
>
> Sincerely,
>
> Paper1846 Authors

---

> ### Author Response · Authors · 2021-08-24
> **Summary of Response to Reviewer vw9x**
>
> Hi Reviewer vw9x
>
> Thanks again for your detailed comments. We hope that the rebuttal, which includes clarification on storing the number of replays, experiments with varying batch sizes, and experiments with optimal edits, adequately addresses your concerns. Please feel free to let us know of any other comments, and we would be happy to address them in our rebuttal and the final version of our paper.
>
> Sincerely,
>
> Paper1846 Authors

---

> ### Author Response · Authors · 2021-08-30
> **Please check on our responses**
>
> Hi Reviewer vw9x
>
> Thanks again for your thoughtful comments and valuable suggestion!
>
> We conducted additional experiments and analysis per your suggestion, to help address the questions raised. This includes clarification on storing the number of replays, experiments with varying batch sizes, and experiments with optimal edits. We hope that our response adequately addresses your concerns. Please feel free to let us know of any other comments, and we would be happy to address them in our rebuttal and the final version of our paper.
>
> Sincerely,

---

### Official Review · Reviewer_BUX3 · 2021-07-17

**Rating:** 6
**Confidence:** 4

**Summary:**

In this paper, the authors propose a technique for replay-based continual learning that is applicable to both task-aware and task-free settings.
Instead of replaying the same memory over and over, each sample from the memory is updated to maximize the forgetting (the increase of loss) through gradient ascent.
Its effectiveness is backed up by extensive experiments.

**Limitations And Societal Impact:**

The performance improvement is rather small.

**Main Review:**

### Weak connection to task-free continual learning
The proposed method is indeed applicable to task-free continual learning.
However, the main technique itself is not a necessary component for task-free continual learning.
I think it is better to introduce it as a general technique that can cover both task-aware and task-free CL.

### Extensive experiments
The authors tested multiple datasets and baselines with repeated experiments.
All numbers are reported with error bars.
Various ablation studies are also the strength of this paper.

### Somewhat marginal improvements
According to the reported scores, the proposed GMED consistently outperforms the baselines.
However, the improvements seem rather small compared to added complexity.

### Lack of deeper insights
GMED performs better than other techniques, but there is not enough analysis about the reasons.
Although I do not think this is a critical flaw but hope there are some explanations or hypotheses.

### Summary
This paper proposes a simple technique that can improve replay-based CL methods.
The idea is not groundbreaking but has simple and understandable intuition behind it.
Although the performance improvements are rather small, the experiments are thorough enough to show the efficacy of the technique.
Overall, I think this is a borderline paper, but I am more on the side of accepting it since there is no significant issue.

**Time Spent Reviewing:**

4

---

> ### Author Response · Authors · 2021-08-10
> **Response to Reviewer BUX3**
>
> We thank the reviewer for the positive review and thoughtful comments!
>
> ### Q1: The improvements are rather small compared to the added complexity.
> We show the computational overhead of GMED is close to ER-MIR [1]. As noted in L371 of the paper: ER-GMED introduces 3 forward and 2 backward passes. ER-MIR introduces 3 forward passes and only 1 backward pass, but 2 of the forward passes are over a much larger set (around 5 times of the batch size) of retrieval candidates. Further, in Table 1, we notice ER-GMED could outperform ER-MIR over Split MNIST and the three most challenging datasets (Split CIFAR-10, Split CIFAR-100, Split mini-ImageNet). Summary of the results are noted under general response in [NewTable 1].
>
> In addition, GMED is also computationally more efficient compared to generative replay based approaches (see the summaries of execution times under the “general response” in [New Table 2]). These approaches simultaneously learn a generative model over data, which requires one additional forward and backward pass over a generative model (which is typically more complicated than the classification model), and an additional forward pass to generate examples to replay; for GEN-MIR, it further introduces overhead to retrieve interfered examples from generated examples. By comparing the official results of GEN-MIR (82.1 on Split MNIST, 80.4 on Permuted MNIST) and ER+GMED (82.67 on Split MNIST and 78.86 on Permuted MNIST). In practice, GEN-MIR takes much longer to run. Therefore, we believe the improvement of ER+GMED is worth extra computational complexity added.
>
> [1] Aljundi et al. Online Continual Learning with Maximally Interfered Retrieval, NeurIPS 2019
>
>
>
> ### Q2: Analysis about the reasons of the improvement of GMED
> The motivation of GMED, similar to previous works on MIR, is based on the hypothesis that interfered examples are more helpful for retraining past performance. There has also been empirical study supporting the hypothesis [2].
>
> In our paper, we additionally presented a hypothesis that the closer Alg+GMED is to optimal gradients, the better the improvements compared to using only Alg. Recall that optimal gradients are computed assuming we have access to all past samples which is not reasonable in online continual learning setup. To this end, in Sec. 4.4 (L309) we present a preliminary analysis to validate the above hypothesis. As such, our analysis shows that the performance improvement brought by GMED is more significant when the editing direction of GMED is more consistent with the optimal editing direction. However, we didn’t find any strong parametric trends.
>
> We agree with the reviewer that our paper doesn’t delve into the exact reasons for the improvements by GMED theoretically. We hope that future work would find our analysis beneficial, and hypothesize that with additional assumptions on the data and the data-stream, one could derive a bound for the gap between GMED edits and optimal edits. We would be happy to include any additional analysis recommended by the reviewer.
>
> [2] Toneva et al. An empirical study of example forgetting during deep neural network learning, ICLR 2019
>
>
> ### Q3: The main technique itself is not a necessary component for task-free continual learning. It is better to introduce it as a general technique that can cover both task-aware and task-free continual learning methods.
> Thank you for the suggestion! The reviewer correctly points that our approach could be applied to both task-aware and task-free setups. We introduced GMED as a “task-free” CL approach as it does not make explicit use of information of task boundaries or task identity information --- this allows the approach to be applied to a broader set of use cases (which are also more challenging).
>
> To validate whether GMED can also benefit task-aware CL methods, we have compared GMED to task-aware approaches in Table 10 in Appendix H. We want to emphasize that our current implementation of GMED does not leverage any information about the task boundaries or task identity, even though both are available in the task-aware setup. As can be seen in Table 10 in Appendix H, integrating GMED to GEM didn’t lead to any significant improvement. We hypothesize that integrating GMED with task-aware setup may require further investigation (e.g., how to make use of the task information to conduct more effective editing), which we leave for future work.

---

> > ### Comment · Reviewer_BUX3 · 2021-08-31
> > **Final score**
> >
> > Thank you for the detailed response.
> > I will maintain my original score.
> >
> > By the way, I still think the proposed method is orthogonal to task-aware/task-free distinction.
> > It is not a critical issue, but introducing it as a task-free CL method seems somewhat misleading.

---

> > > ### Author Response · Authors · 2021-09-02
> > > **Thanks for the replies**
> > >
> > > Hi Reviewer BUX3
> > >
> > > Thank you for the suggestion. We agree with this idea that GMED is not tied to whether task information is available or not. In our final version of the paper, we would update the writing to avoid such confusion.
> > >
> > > Sincerely,
> > >
> > > Paper1846 Authors

---

> ### Author Response · Authors · 2021-08-16
> **Overview of Responses to Reviewer BUX3**
>
> Hi Reviewer BUX3
>
> Thank you again for your detailed review. We are glad that you found our idea intuitive, and our set of experiments extensive enough to show the efficacy of the method. For convenience, please find below an overview of the points covered in the rebuttal:
>
> * Detailed comparison of GMED with previous state-of-art w.r.t. computation overhead and training time (especially significant when compared to GEN-MIR), with the improvements justifying the additional complexity.
> * Revisiting optimal edits experiments which provide some pointers as to why GMED works.
> * Effect of GMED in the task-free and task-aware setups.
>
> We hope that our rebuttal adequately addresses your concerns and questions. Please feel free to provide any additional comments, and we would be happy to address them in the rebuttal as well as in the final version of our paper.
>
> Sincerely,
>
> Paper1846 Authors

---

> ### Author Response · Authors · 2021-08-24
> **Summary of Response to Reviewer BUX3**
>
> Hi Reviewer BUX3
>
> Thanks again for your detailed comments. We hope that the rebuttal, which includes comparison w.r.t computation overhead, experiments with optimal edits, and experiments on task-aware setups, adequately addresses your concerns. Please feel free to let us know of any other comments, and we would be happy to address them in our rebuttal and the final version of our paper.
>
> Sincerely,
>
> Paper1846 Authors

---

> ### Author Response · Authors · 2021-08-30
> **Please check on our response**
>
> Hi Reviewer BUX3
>
> Thanks again for your thoughtful comments and valuable suggestion!
>
> We provide more detailed results and conducted additional analysis per your suggestion, to help address the questions raised. This includes comparison w.r.t computation overhead, experiments with optimal edits, and experiments on task-aware setups. We hope that our response adequately addresses your concerns and that you could reconsider the evaluation of this work.
>
> Please feel free to let us know of any other comments, and we would be happy to address them in our rebuttal and the final version of our paper.
>
> Sincerely,

---

### Author Response · Authors · 2021-08-10
**General Response to the Reviewers**

We thank the reviewers for their positive reviews and helpful comments. We are delighted that the reviewers found GMED to be novel, well-motivated, intuitive and simple to implement. We are glad to find that the reviewers’ found our set of experiments extensive and supporting the effectiveness of GMED.

We very much appreciate the feedback regarding the modest improvements on the absolute scale. We address this point below and respond to the individual reviewers’ comments in their respective sections.

### [Improvements on absolute scale are modest]
A common point raised by the reviewers’ is that improvements brought by GMED, while statistically significant, are marginal on absolute value scale as compared to previous works.

We agree here. However we would like to emphasize, even though on the absolute scale the improvements are modest we do find GMED giving best performance (i.e., new state-of-the-art results) when combined with different CL baselines over a wide range of datasets. Performance comparison with previous SOTA results on each dataset are summarized in the table below. We note that GMED’s improvement over the previous state-of-the-art result on each dataset is statistically significant with p < 0.01 / 0.05 / 0.1.

**New Table 1: Performance comparison between ER+GMED and previous SoTA**


Method/Dataset | Split MNIST | Rotated MNIST | Split CIFAR-10 | Split CIFAR-100 | Split mini-ImageNet
----------------------| ---------------- | --------------------- | -------------------- | ---------------------- | --------------------
GMED               |  86.52±1.4    |    80.61±1.2       |     47.47±3.2     |   21.22±1.0         |    31.81±1.3
Previous SoTA  |  85.72***±1.2    |    79.23*±0.7       |     46.29*±2.7     |   20.02**±1.7          |    30.77*±2.2


\* p < 0.1    ** p < 0.05    *** p < 0.01

Further, we would like to note that the computational overhead (run time for model training) of GMED is close to ER-MIR[1] (sometimes faster), and is much smaller compared to generative replay methods such as MIR-GEN also proposed in [1] . We summarize the wall-clock running time of ER-GMED and its comparison to baselines (ER, ER-MIR, GEN-MIR) in the table below.


**New Table 2: Runtime comparison between ER-GMED and other baselines**


Method/Dataset | Split MNIST | Rotated MNIST | Split CIFAR-10 | Split mini-ImageNet
--------------------- | ---------------- | --------------------- | -------------------- | ---------------------------
ER                     |       6 sec      |         16 sec       |      7 min            |           32 min
ER-MIR             |       7 sec      |         36 sec       |       10min          |          47 min
GEN-MIR*         |       5 min        |           -              |          -               |           -
ER-GMED         |       7 sec      |         31 sec       |       11 min        |           46 min


\* We used the official released code for GEN-MIR for conducting this runtime experiment.

This is in line with our analysis presented in L371 regarding extra forward and backward passes of models: GMED introduces 3 forward and 2 backward passes while ER-MIR introduces 3 forward passes and only 1 backward pass, but 2 of the forward passes are over a much larger set (around 5 times of the batch size) of retrieval candidates.

Finally, we want to emphasize that GMED is a general technique which can complement existing (and we hope future) memory-based continual learning methods. Experiments in our paper show that GMED can complement memory-based CL methods that are performing advanced memory replays (e.g., in MIR) and data augmentation over memory examples (e.g., ER_aug). We believe that GMED can be combined with new memory-based CL methods which conduct more sophisticated memory scheduling (what examples to store in the memory) or memory replays, yielding new state-of-the-art results in the future.

[1] Aljundi et al. Online Continual Learning with Maximally Interfered Retrieval, 2019

---

### Decision · Program_Chairs · 2021-09-27

**Decision:**

Accept (Poster)

**Comment:**

This paper introduces the Gradient based Memory Editing (GMED) framework for task-free online continual learning that edits stored examples using gradient updates. The paper is well-written. The proposed method is interesting, novel, and has simple understandable intuition, though theoretical justifications are lacking. Experiments are extensive. While improvements are somewhat marginal, the experiments are thorough enough to show efficacy of the technique.